# OpenReview forum: "GuardAgent: Safeguard LLM Agents via Knowledge-Enabled Reasoning"
_ICML.cc/2025/Conference — ICML 2025 poster_

### Official Review · Reviewer_WcSs · 2025-03-13

**Overall Recommendation:** 4

**Summary:**

This paper proposes GuardAgent, a new framework designed to safeguard LLM agents by checking if their actions satisfy specific safety guard requests. GuardAgent has two main steps: 1) analyzes safety guard requests and generates a task plan. 2) Then, convert this plan into the code and execute it. The authors further develop two benchmarks for evaluating  GuardAgent: EICU-AC (design for healthcare agents) and Mind2Web-SC (design for web agents). Experiments show that GuardAgent achieves 98% guardrail accuracy on EICU-AC and 83%-90% on Mind2Web-SC across several LLMs (GPT-4, Llama3-70B, and Llama3.1-70B).

**Claims And Evidence:**

Yes

**Essential References Not Discussed:**

[1] TrustAgent: Towards Safe and Trustworthy LLM-based Agents through Agent Constitution. ICML 2024 Workshop TiFA.

**Experimental Designs Or Analyses:**

Experimental designs are reasonable. The author provides benchmark, dataset and metrics.

**Methods And Evaluation Criteria:**

Yes

**Other Comments Or Suggestions:**

N/A

**Other Strengths And Weaknesses:**

Strengths:
1. Comprehensive system design: GuardAgent involves several necessary LLM-based agent components (planning, code generation, memory). The code-based approach provides more reliable guardrails compared to natural language-based alternatives.

2. Comprehensive experiment results: The authors developed two benchmarks (healthcare and web agent) and evaluated their approach across multiple backbone LLMs, demonstrating consistent performance. The paper also includes thorough comparisons with baseline approaches, including "model guarding agents" (using LLMs with carefully crafted prompts) and "invasive" guardrails (hardcoded into agent systems).

3. Good writing and easy to follow.

Weakness:
1.  The application scope only focuses on healthcare applications and web navigation. In related work, the author mentioned several LLM-based agent applications, such as finance, autonomous driving, and education. The generalizability of GuardAgent needs to be further explored.

2. GuardAgent’s performance improvement relies on in-context learning and memory components. The paper doesn't explore how the quality of these demonstrations affects performance, though it does show that performance degrades with less relevant demonstrations.

3. While the author mentions a debugging mechanism, the details and effectiveness of this component aren't thoroughly explored. Could the author provide more evidence or discussion of the "debugging" component?

**Questions For Authors:**

See in weakness.

**Relation To Broader Scientific Literature:**

This paper makes a reasonable contribution to the field of LLM-based agent safety. Although this paper only discusses "healthcare agent" and "web agent". Other agents may be able to draw on the experimental design ideas and evaluation criteria in this work.

**Theoretical Claims:**

This work focuses more on experimental design and benchmarking. There is not much Theoretical. But I carefully read all section 4 "GuardAgent Framework".

---

> ### Author Rebuttal · Authors · 2025-04-01
>
> Thank you for reviewing our paper and your positive feedback on our contributions. Please find our detailed responses to your comments below.
>
> **W1: Generalizability of GuardAgent needs to be further explored**
>
> **A1**: Thank you for the comment. In Appendix P, we have included an application of GuardAgent beyond healthcare and web navigation: common sense reasoning, where GuardAgent outperforms the baseline in predicting the risks of task execution.
>
> Here, we add another experiment to demonstrate GuardAgent’s efficacy in safeguarding scientific reasoning tasks. We use the MMLU dataset, which contains questions from 57 different subjects. We divide these subjects into four major categories: Mathematics and Logic, Natural Science, Social Science, and Technology and Engineering. The target agent is designed to answer questions from these categories, with the requirement that the user must possess expertise in the relevant category. Note that this setting simulates access control in Q&A.
>
> GuardAgent receives a question along with the user’s expertise. If the question aligns with the user’s area of expertise, the target agent is allowed to respond. GuardAgent achieves a 100% success rate in Label Prediction Accuracy (LPA, measuring the overall correctness of the safeguard) with GPT-4. We will include this experiment in the revised paper. Thank you again for your suggestion.
>
> **W2: How the quality of the demonstrations affects performance**
>
> **A2**: Thank you for the comment. As you mentioned, the performance of GuardAgent is not significantly degraded with fewer demonstrations.
>
> Regarding the quality of the demonstrations, we have included an experiment in Appendix M, where we retrieve demonstrations based on "least-similarity" instead of "max-similarity". As shown in Table 5 of our paper, the safeguard accuracy of GuardAgent drops from 98.7% to 98.1% on EICU-AC and from 90.0% to 84.0% on Mind2Web-SC. The results indicate that reducing the relevance of the retrieved memories only causes moderate degradation in the performance of GuardAgent.
>
> Following your suggestion, we present another experiment to further explore the impact of demonstration quality on GuardAgent’s performance. In the default setting, only correct safeguard examples are added to the memory base. Here, we modify the setting by storing all executions GuardAgent indiscriminately. In the table below, we observe that there is a less than 9% drop in absolute safeguard accuracy across all configurations compared with the results in our Table 1. These results highlight the decent robustness of GuardAgent to the quality of the demonstrations.
> ||EICU-AC|||||Mind2Web-SC|||||
> |-|-|-|-|-|-|-|-|-|-|-|
> ||LPA|LPP|LPR|EA|FRA|LPA|LPP|LPR|EA|FRA|
> |llama3|93.0|100|86.4|86.4|100|81.5|98.5|65.0|64.0|100|
> |llama3.1|91.5|100|86.8|79.9|100|83.0|92.3|72.0|72.0|100|
> |gpt-4|90.19|90.2|91.3|87.6|89.6|81.5|89.0|73.0|72.0|100|
>
> Please note that in practice, an evaluator (e.g., human feedback) can be employed to determine which agent executions should be added to the memory. Such evaluation ensures the quality of future retrieved demonstrations.
>
> Thank you again for the suggestion!
>
> **W3: More evidence or discussion of the "debugging" component**
>
> **A3**: Thank you for the question. The debugging component prompts the core LLM with the guardrail task, the generated guardrail code, the errors raised from code execution, and a request to regenerate the guardrail code.
>
>
> We have added a detailed discussion of the debugging mechanism in Appendix K. In most cases, debugging is not activated; therefore, we created a more challenging scenario by removing both the toolbox and memory from GuardAgent. Consequently, 29 out of 316 generated codes were not executable initially, including 11 name errors, 3 syntax errors, and 15 type errors. Debugging resolved 9 of these 29 errors, specifically 8 name errors and 1 type error. None of the syntax errors were successfully debugged; they were all caused by the incorrect representation of the newline symbol as '\\\n'.
>
>
> We will include a complete debugging prompt and the corresponding correction of the guardrail code in our revised paper. Thank you for your suggestion.
>
> **W4: Missing reference**
>
> **A4**: Thank you for bringing this important work to our attention. TrustAgent develops and implements an agent constitution to ensure the safety of actions and tool utilization of the target agent. This work is highly relevant to ours, and we will definitely include a discussion about it in our revised paper.
>
>
> Thank you once again for your valuable feedback. We are happy to answer your follow-up questions if there are any.

---

> > ### Comment · Reviewer_WcSs · 2025-04-03
> >
> > Thanks to the author's rebuttal. It addresses part of my concerns. I have raised my score.

---

> > > ### Author Response · Authors · 2025-04-03
> > >
> > > Thank you for your acknowledgement of our efforts and for raising the score. And thank you again for reviewing our paper.

---

### Official Review · Reviewer_KT5h · 2025-03-14

**Overall Recommendation:** 3

**Summary:**

This paper proposes a framework that safeguards LLM agents by using an agent (GuardAgent) to check whether their actions comply with safety requirements. GuardAgent uses a two-step process: generating a task plan based on safety requirements, then converting this plan into executable guardrail code. The authors introduce two benchmarks (EICU-AC for healthcare access control and Mind2Web-SC for web safety) and demonstrate effectiveness with 98% and 83% guardrail accuracies respectively.

**Claims And Evidence:**

The claims about guardrail accuracy and flexibility are supported by experimental results across multiple LLM backbones. However, the "low operational overhead" claim lacks comprehensive analysis against alternatives.

**Essential References Not Discussed:**

- More discussion of competing frameworks for agent guardrails like Langchain's guardrails library would provide better context on the landscape of agent safety approaches.
- References to formal verification approaches for LLM outputs, which could complement the code-based guardrails proposed in GuardAgent.

**Experimental Designs Or Analyses:**

Some notable strengths:
- The comparison against multiple baselines, including both "model guarding agents" and invasive approaches directly modifying the target agent's prompt.
- Comprehensive ablation studies examining the impact of memory size and toolbox components.
- Detailed breakdown of performance by role in EICU-AC and rule in Mind2Web-SC to identify potential weaknesses.
- Testing with multiple LLM backbones to demonstrate robustness across different models.

Limitations:
- It does not provide analyses on false refusals where benign actions has been wrongly denied

**Methods And Evaluation Criteria:**

The evaluation metrics (LPA, LPP, LPR, EA, FRA) appropriately capture both guardrail effectiveness and impact on target agent functionality. However, the benchmarks are specifically designed for narrow domains, which may not necessarily represent a broad spectrum of real-life situations.

**Other Comments Or Suggestions:**

N/A

**Other Strengths And Weaknesses:**

This paper is looking at an important problem, which is a plus.

**Questions For Authors:**

- The guardagent framework assumes that safety specifications are available, however exhaustive specifications are not always feasible. How would a guardagent perform in that scenario?
- Do you observe false refusals and how often?
- Is code generation and execution always feasible in a general setting?

**Relation To Broader Scientific Literature:**

- They distinguish between "model guarding models" approaches (like LlamaGuard) and their "agent guarding agents" approach, clearly explaining why traditional guardrails designed for textual content are insufficient for agent actions.

- They acknowledge and build upon existing work on knowledge-enabled reasoning in LLM agents, particularly the use of in-context demonstrations and retrieval mechanisms.

- They contextualize their work within the evolving landscape of safety concerns for AI agents, citing relevant literature on potential misuse scenarios.

**Theoretical Claims:**

The paper doesn't make formal theoretical claims requiring mathematical proofs. The claims are purely empirical.

---

> ### Author Rebuttal · Authors · 2025-04-01
>
> Thank you for reviewing our paper and your positive feedback. Please find our responses to your comments below:
>
> **W1: The “low operational overhead” claim**
>
> **A1**: We apologize for the ambiguity. The “low operational overhead” refers to one of the three key advantages of GuardAgent – it “employs the core LLM by in-context learning, enabling direct utilization of off-the-shelf LLMs without the need for additional training” (line 106-108 left). Therefore, compared with trained guardrails like LlamaGuard, our method has low operational overhead. We will change the statement by directly saying that GuardAgent is free of training.
>
> Regarding the empirical time cost, we have included Table 6 in the appendix. Both GuardAgent and the “model-guarding-agent” baseline are free of training – they both achieve the same level of time cost as the target agent.
>
> Thank you for pointing this out.
>
> **W2: The benchmarks are specifically designed for narrow domains**
>
> **A2**: Thank you for the comment. The healthcare domain (and code generation) covered by EICU-AC and the web applications covered by Mind2Web-SC are two important fields for LLM agent applications. In Appendix P, we also curated a dataset from CSQA to evaluate GuardAgent on common knowledge QA with certain rules. In our future work, we will create a complex dataset combining the ones we have proposed in the paper and more data for diverse application domains to better cover a broader spectrum of real-life situations.
>
> **W3&Q2: Analyses on false refusals**
>
> **A3**: Thank you for your constructive suggestion. In the table below, we show the false refusal rate of GuardAgent compared with the model-guarding-agent baseline. GuardAgent achieves close to zero false refusal rates in all configurations except on Mind2Web-SC with Llama3.1-70B.
>
> ||EICU-AC|||Mind2Web-SC|||
> |-|-|-|-|-|-|-|
> ||Llama3-70B|Llama3.1-70B|GPT-4|Llama3-70B|Llama3.1-70B|GPT-4|
> |Baseline|4.5|2.6|5.2|4.0|3.0|0|
> |GuardAgent|0|0|0|1.0|14.0|0|
>
> We examined the false refusal cases for this configuration and found that most false refusals were caused by incorrect associations between the agent's actions and the safety rules. For example, when the target agent is instructed to "Search for running shoes," GuardAgent incorrectly applies the rule for "Search/Find/Show movies/music/video," while the correct rule is for "Shopping." This error likely resulted from the exact word "Search" matching in both the user query and the incorrect rule.
> We will include these results in our revised paper. Thank you for your suggestion!
>
> **W4: More discussion of 1) competing frameworks for agent guardrails and 2) formal verification approaches**
>
> **A4**: Thank you for pointing out these important related directions. The agent guardrails, such as the guardrail libraries for LangChain and LlamaIndex, belong to the model-guarding-agent family, where the guardrail models are specifically trained to detect particular categories of harmfulness. Formal verification is another important type of approach to ensure the safety of LLM outputs. For example, C-RAG provides theoretical guarantees on the generation risk of LLMs with RAG [1]; FVEL provides formal verification for LLM-generated code by transforming it into a verifiable environment [2].
>
> We will add a more detailed discussion with extended references in our revised paper. Thank you again for the advice.
>
> [1] Kang et al. C-RAG: Certified Generation Risks for Retrieval-Augmented Language Models, ICML 2024.
> [2] Lin et al. FVEL: Interactive Formal Verification Environment with Large Language Models via Theorem Proving, NeurIPS 2024.
>
> **Q1: What if safety specifications are unavailable?**
>
> **A5**: Thank you for the insightful question. GuardAgent is designed to follow user-desired safety rules. When there are no user-provided rules, GuardAgent could incorporate external knowledge by calling a searching API to establish a “constitution”. For example, when the target agent is designed for drug development, FDA regulations will be recommended with high probability. We will mention these potential extensions to GuardAgent in the discussion of future work.
>
> **Q3: Is code generation and execution always feasible in a general setting?**
>
> **A6**: Thank you for the insightful question. Based on our evaluation in Appendix P, GuardAgent can handle most text-based tasks using code generation and execution, which covers a substantial of the task space of LLMs and agents.
>
> One possible explanation is that most rules can be abstracted into decision trees, where the inputs are the action trajectories of the target agent and the output indicates whether a rule has been violated or followed. This structure makes decision-making through code generation and execution feasible in many cases.
>
> We will add the discussion above to support the design of code generation and execution. Thank you for the question.
>
> Thank you for your valuable comments. Please let us know if you have any follow-up questions.

---

### Official Review · Reviewer_BFcf · 2025-03-14

**Overall Recommendation:** 3

**Summary:**

This paper proposes GuardAgent, a novel framework to safeguard LLM agents by leveraging knowledge-enabled reasoning. The approach involves a two-step process where an LLM generates a detailed task plan from safety guard requests and then produces executable guardrail code via in-context learning with retrieved demonstrations. The authors introduce two benchmarks—EICU-AC for healthcare access control and Mind2Web-SC for web agent safety control—to evaluate the method. Experimental results indicate that GuardAgent significantly outperforms baseline methods (including models with well-crafted prompts and a model-guarding baseline) on multiple metrics (label prediction accuracy, explanation accuracy, etc.) without impairing the underlying agent’s task performance.

**Claims And Evidence:**

The paper claims that GuardAgent provides flexible and reliable guardrails for diverse LLM agents, improving upon baseline approaches by converting safety requests into executable code. While the experimental evidence supports high accuracy on the provided benchmarks, several claims remain partially unsubstantiated:

- It is unclear how well the generated guardrails will hold up against dynamically evolving attack surfaces in real-world settings.

- The experiments do not assess the consistency of guardrail outputs across multiple runs, which is crucial for reliable defense.

**Essential References Not Discussed:**

While the paper cites a solid set of references, it would be helpful to see comparisons with recent works that employ multi-agent setups for risk assessment or those that specifically address dynamic adversarial environments. For instance, referencing recent advances in adversarial robustness for LLM agents or discussing related work on using a second LLM as a judge for potential risks would further situate the contribution. For example: Hua, Wenyue, et al. "Trustagent: Towards safe and trustworthy llm-based agents through agent constitution." Trustworthy Multi-modal Foundation Models and AI Agents (TiFA). 2024.

**Experimental Designs Or Analyses:**

The experimental design is thorough in terms of benchmark creation and comparison with a well-crafted baseline and an invasive guardrail approach. Nonetheless, the evaluation could be strengthened by:

- Including additional state-of-the-art LLMs, especially those known for strong reasoning capabilities (e.g., OpenAI o1 and DeepSeek R-1).

- Considering baselines that use an LLM agent as a judge for attack and risk assessment, which might offer a stronger point of comparison.

- Demonstrating defense performance against more sophisticated and adversarial attack scenarios to assess scalability and generalizability.

**Methods And Evaluation Criteria:**

The methodology, focusing on task planning and code generation, is interesting and well-motivated. The use of in-context demonstrations to bridge guard requests and executable checks seems to be effective. However, the evaluation would benefit from:

- Additional metrics that capture the consistency and robustness of the guardrails over repeated executions.

- Experiments addressing how the method adapts to dynamically changing attack surfaces.

**Other Comments Or Suggestions:**

- Consider discussing potential strategies for updating or adapting guardrails in real-time as new attack vectors emerge.

- Including a sensitivity analysis of the in-context retrieval process could shed light on the robustness of the approach.

- A deeper discussion on scalability in more complex, multi-faceted scenarios would be beneficial.

**Other Strengths And Weaknesses:**

Strengths:

- Innovative use of in-context learning to generate code-based guardrails.

- Clear empirical improvements over baseline methods on two benchmarks.

- Non-invasive design that preserves target agent performance.


Weaknesses:

- Limited evaluation on how the approach handles dynamically changing attack surfaces.

- Lack of metrics or analysis on the consistency of the generated guardrails.

- Evaluation restricted to a small set of LLMs; more comparisons with cutting-edge models (e.g., OpenAI o1, DeepSeek R-1) would strengthen the paper.

- Baseline comparisons could be extended to include approaches that use an additional LLM agent as a judge for potential attacks.

**Questions For Authors:**

- How does GuardAgent adapt or update its guardrail code when facing dynamically changing attack surfaces over time?

- Can you provide additional experiments or metrics that assess the consistency of the generated guardrails across multiple runs?

- Have you considered evaluating GuardAgent with additional state-of-the-art LLMs, such as OpenAI o1 and DeepSeek R-1, to further validate its performance?

- Could you elaborate on how the framework would perform under more sophisticated adversarial scenarios and whether it can scale to cover such cases?

**Relation To Broader Scientific Literature:**

This work builds upon and extends ideas from recent works in LLM moderation (e.g., LlamaGuard) and agent-based reasoning. The integration of code generation for safety enforcement is a promising direction that aligns with the growing literature on using structured reasoning for reliability in AI systems.

**Theoretical Claims:**

The paper does not present detailed theoretical proofs; rather, it focuses on empirical validation of the proposed framework. The high-level conceptual reasoning appears sound, but the absence of theoretical guarantees regarding robustness against evolving attacks leaves some questions unanswered.

---

> ### Author Rebuttal · Authors · 2025-04-01
>
> Thank you for reviewing our paper and your positive feedback. We especially thank you for recognizing the importance of our work and our contributions. Please find our responses to your comments below.
>
> **W1&Q1&W5: Dynamically evolving attack surfaces and more sophisticated adversarial scenarios**
>
> **A1**: Thank you for the insightful question. Our GuardAgent is designed to effectively follow safety rules, such as access control requirements systematically, by generating corresponding codes and executing them; therefore it is flexible to integrate additional safety rules, such as the ones that could describe the potential attack strategies as "constitutions" to make GuardAgent more and more resilient over time by identifying more unknown adversarial strategies. With that being said, it is possible to enhance the resilience of GuardAgent continuously by periodically summarizing the uncovered new adversarial strategies, and this framework will be more efficient than traditional methods such as adversarial training! We will add such discussion in the paper following your suggestions and we believe this will lead to a series of interesting new research work!
>
> **W2&Q2: Consistency of guardrail outputs across multiple runs**
>
> **A2**: Thank you for your suggestion. We add a new experiment to show the consistency of GuardAgent over multiple runs. We test GuardAgent 5 times on the two datasets and show the percentage of examples where GuardAgent made 5, 4, 3, 2, or 1 correct predictions. For most examples, GuardAgent gave similar results in the 5 runs (with at least 4 out of 5 or no correct predictions). We will include this experiment in the revised paper. Thank you!
> |||5|4|3|2|1|0|
> |-|-|-|-|-|-|-|-|
> |EICU-AC|llama3|83.9|12.3|1.2|2.2|0.3|0|
> ||llama3.1|83.5|15.1|0.3|0.3|0.6|0|
> |Mind2Web-SC|llama3|80.5|1.0|1.5|0|1.5|15.5|
> ||llama3.1|71.0|13.0|3.0|1.5|3.5|8.0|
>
> **W3&Q3: Test on additional state-of-the-art LLMs**
>
> **A3**: Thank you for the valuable suggestion. Please find the evaluation results for GuardAgent on o1 and r1 in the table below:
> ||EICU-AC|||||Mind2Web-SC|||||
> |-|-|-|-|-|-|-|-|-|-|-|
> ||LPA|LPP|LPR|EA|FRA|LPA|LPP|LPR|EA|FRA|
> |r1|99.7|100|100|87.7|100|85.5|92.8|77.0|77.0|100|
> |o1|100|100|100|98.3|100|87.5|95.2|79.0|76.0|100|
>
> GuardAgent performs well (compared to the results in Table 1 of our paper) on these reasoning LLMs. We will add the results to our revised paper. Thank you for your suggestion!
>
> **W4&Q4: Comparison with LLM agent as a judge for risk assessment, e.g., TrustAgent.**
>
> **A4**: Thank you for bringing this important work to our attention! TrustAgent is indeed relevant to our work as it safeguards LLM agents based on constitutions established for diverse agent application domains. Below, we compare GuardAgent with TrustAgent, both with GPT-4.
> ||EHRAgent+EICU-AC|||SeeAct+Mind2Web-SC|||
> |-|-|-|-|-|-|-|
> ||Accuracy|Precision|Recall|Accuracy|Precision|Recall|
> |GuardAgent|98.7|100|97.5|90.0|100|80.0|
> |TrustAgent|52.8|53.9|63.7|47.0|47.5|56.0|
>
> Here, we focus on the risk assessment of TrustAgent and combine its “risky” categories into a single label to better fit the binary classification setting. Moreover, we replaced the original constitution of TrustAgent with the safety requests used in our work, which are more compatible with the two agents here. The results indicate that TrustAgent cannot adequately handle these safety requests. We discovered that the primary reason is that TrustAgent is designed to adhere to a general concept of safety, aimed at safeguarding textual-based agent *planning*. Conversely, GuardAgent verifies whether the agent's *execution process* (such as the code generated by EHRAgent) adheres to the established safety rules. Thank you for your comment!
>
> **W6: Sensitivity analysis of the in-context retrieval process**
>
> **A5**: Thank you for this constructive suggestion. We have included two sensitivity analyses regarding the in-context retrieval process. First, in Section 5.3, we found that GuardAgent still performs well with fewer demonstrations. Second, in Appendix M, we discovered that memory retrieval based on "least-similarity" instead of "max-similarity" results in only moderate degradation to GuardAgent’s performance.
>
> Here, we add an experiment to examine the impact of the quality of retrieved demonstrations. Instead of storing only the correct executions of GuardAgent in the memory bank, we store all executions indiscriminately.
> ||EICU-AC|||||Mind2Web-SC|||||
> |-|-|-|-|-|-|-|-|-|-|-|
> ||LPA|LPP|LPR|EA|FRA|LPA|LPP|LPR|EA|FRA|
> |llama3|93.0|100|86.4|86.4|100|81.5|98.5|65.0|64.0|100|
> |llama3.1|91.5|100|86.8|79.9|100|83.0|92.3|72.0|72.0|100|
> |gpt-4|90.2|90.2|91.3|87.6|89.6|81.5|89.0|73.0|72.0|100|
>
> Compared with Table 1 of our paper, there is a less than 9% drop in absolute safeguard accuracy across all configurations, demonstrating decent robustness of GuardAgent to the quality of the demonstrations.
>
> Please let us know if you have further questions. Thank you again!

---

### Official Review · Reviewer_aMr9 · 2025-03-17

**Overall Recommendation:** 3

**Summary:**

GuardAgent is the first guardrail agent designed to monitor and regulate the actions of LLM agents. It operates by leveraging LLMs to translate security requirements into executable guardrail code. A memory module is utilized to enhance guardrail performance by retrieving past task demonstrations. Experimental results demonstrate high accuracy on two benchmarks: EICU-AC (for healthcare access control) and Mind2Web-SC (for web agent safety).

**Claims And Evidence:**

N/A

**Essential References Not Discussed:**

N/A

**Experimental Designs Or Analyses:**

N/A

**Methods And Evaluation Criteria:**

N/A

**Other Comments Or Suggestions:**

N/A

**Other Strengths And Weaknesses:**

Strengths
- Ensuring security and safety in LLM systems is crucial in today's AI landscape, and this paper addresses a highly relevant issue.
- The study presents practical scenarios, including EICU-AC (Healthcare agent) and the Mind2Web-SC (Mind2Web-Safety Control) benchmark, to evaluate the proposed approach.

Weaknesses
- The authors heavily rely on LLM capabilities to build a safety-guarding agent. However, the approach presented in Section 4 appears to be more of an engineering implementation rather than a novel conceptual advancement. This makes the paper feel more like a technical report than a research contribution.
- The paper does not introduce any novel methodology to address security and safety challenges in LLMs.
- Additionally, the experimental results do not seem to reveal any significant new insights or discoveries.

**Questions For Authors:**

N/A

**Relation To Broader Scientific Literature:**

minor contribution to scientific literature

**Theoretical Claims:**

N/A

---

> ### Author Rebuttal · Authors · 2025-04-01
>
> Thank you for reviewing our paper and your positive ratings! We are glad that you acknowledge the importance of the problem we are solving and the practicality of our settings. In the following, we reply to your concerns one by one.
>
> **W1: Novel conceptual advancement**
>
> **A1**: Thank you for your valuable comment. Our work introduces several key conceptual advancements, including the following:
>
> - Prior to our work, guardrails for LLMs typically referred to trained models that detect whether a target LLM’s inputs and/or outputs are harmful. We extend this concept by introducing guardrails for LLM agents that monitor whether an agent's actions comply with prescribed rules.
> - Existing guardrails for LLMs are model-based. In contrast, our approach demonstrates the advantages of using an LLM agent, rather than a standalone model, to safeguard other LLM agents.
> - Our work highlights the effectiveness of a reasoning-then-action pipeline – consisting of task planning followed by code generation and execution – augmented by knowledge retrieval from memory, in enhancing the safety of LLM agents.
> These conceptual contributions underscore both the novelty and the significance of our work.
>
> **W2: Novel methodology to address security and safety challenges in LLMs**
>
> **A2**: Thank you for your comment on our methodology. Our approach incorporates several key innovations:
> - Our guardrail mechanism is based on code generation and execution, providing strong reliability since all decision outcomes depend on the successful execution of correctly generated code.
> - We employ in-context learning for task planning and code generation, eliminating the need for model training.
> - We introduce an extensible toolbox for storing supportive functions used in code generation, which enhances GuardAgent’s flexibility in handling novel guardrail requests.
> These design choices have been empirically validated to be effective in achieving their intended goals, as demonstrated by our experimental results.
>
> **W3: Significant new insights or discoveries in the experimental results**
>
> **A3**: Thank you for your constructive comments. In our revised paper, we will highlight the following key insights derived from our experiments:
>
> - The comparison between our code-based guardrail and the text-based guardrail (i.e., the baseline) demonstrates the superior reliability of code-based guardrails in safeguarding LLM agents.
> - Our evaluation of GuardAgent on the two created benchmarks (EICU-AC and Mind2Web-SC) and the commonsense reasoning dataset CSQA underscores the generalizability of code-based guardrails.
> - Interestingly, we observed that LLM agents tend to generate code-based guardrails even when not explicitly instructed to do so. A potential explanation is that safety rules are often naturally represented as decision trees, where inputs consist of the target agent’s action trajectories and outputs indicate whether a rule has been followed or violated. This structure aligns well with a code generation and execution paradigm, making it a more suitable approach for safety decision-making in many scenarios.
>
> Thank you again for your valuable comments! We are happy to address your remaining concerns if there are any.

---

### Decision · Program_Chairs · 2025-05-01

**Decision:**

Accept (poster)

**Comment:**

This paper studies systems where one LLM acts as a "guard" to prevent another LLM from doing unsafe things. The guard LLM uses in-context learning to write code to allow or deny the other LLM's requests for data access or permission to take actions. All reviewers agreed the topic was important and timely, though the results may not have been too surprising. The reviewers appreciated that this paper included "comprehensive" tests, which they ran in several different contexts, and using several different LLMs -- both with and without "reasoning" capabilities, the latter added in the rebuttal phase. No major issues emerged during the discussion phase and at least one reviewer raised their score after the discussion.